# Therapeutic Drug-Induced Metabolic Reprogramming in Glioblastoma

**DOI:** 10.3390/cells11192956

**Published:** 2022-09-22

**Authors:** Trang T. T. Nguyen, Enyuan Shang, Mike-Andrew Westhoff, Georg Karpel-Massler, Markus D. Siegelin

**Affiliations:** 1Department of Pathology and Cell Biology, Columbia University Medical Center, New York, NY 10032, USA; 2Department of Biological Sciences, Bronx Community College, City University of New York, Bronx, NY 10453, USA; 3Department of Pediatrics and Adolescent Medicine, Ulm University Medical Center, 89075 Ulm, Germany; 4Department of Neurosurgery, Ulm University Medical Center, 89081 Ulm, Germany

**Keywords:** glioblastoma, metabolism, glycolysis, TCA cycle, oxidative phosphorylation (OXPHOS)

## Abstract

Glioblastoma WHO IV (GBM), the most common primary brain tumor in adults, is a heterogenous malignancy that displays a reprogrammed metabolism with various fuel sources at its disposal. Tumor cells primarily appear to consume glucose to entertain their anabolic and catabolic metabolism. While less effective for energy production, aerobic glycolysis (Warburg effect) is an effective means to drive biosynthesis of critical molecules required for relentless growth and resistance to cell death. Targeting the Warburg effect may be an effective venue for cancer treatment. However, past and recent evidence highlight that this approach may be limited in scope because GBM cells possess metabolic plasticity that allows them to harness other substrates, which include but are not limited to, fatty acids, amino acids, lactate, and acetate. Here, we review recent key findings in the literature that highlight that GBM cells substantially reprogram their metabolism upon therapy. These studies suggest that blocking glycolysis will yield a concomitant reactivation of oxidative energy pathways and most dominantly beta-oxidation of fatty acids.

## 1. Introduction

To date, metabolism has not been extensively studied in glial tumors. Glioblastoma WHO IV (GBM) constitutes a significant health care challenge because for many decades the prognosis has not significantly changed [1,2]. Since it is a primary brain tumor (unlike the brain metastasis), it originates from the glial cells in the brain, e.g., astrocytes. In part because of its origination from glial cells, GBM cells display a rather diffuse infiltrative margin and by no means it is feasible to resect every single tumor cell by surgery. In addition, single GBM cells may be elegantly protected by the blood–brain barrier from certain therapeutic modalities. Moreover, single cell studies have demonstrated a substantial heterogeneity of GBMs, implicating that targeting a single pathway will unlikely be successful in a durable manner [3]. All these and other hallmarks of GBM provide the foundation for a highly unfavorable prognosis of only 12–15 months of overall survival [4,5].

In this review, we will summarize how certain therapies (some of which failed clinical trials) affect GBM metabolism and how such knowledge may be used for the development of more effective therapies. Our focus will predominantly rest on glycolysis, cellular respiration, and fatty acid oxidation.

## 2. Glucose Metabolism as a Central Component of GBM and Tumor Growth

Metabolism constitutes the break-down (catabolism) and biosynthesis (anabolism) of fuel sources. Glucose is the main energy/fuel source for most tissues. Its catabolism is referred to as glycolysis and is initiated by the uptake of glucose into the cells, which is followed by a kinase reaction to activate glucose that is mediated by hexokinase (e.g., muscle and cancer cells) or glucokinase (liver) [6]. Glucose uptake in muscle and adipose tissue is regulated by the GLUT4 transporter (glucose transporter type 4), which depends on insulin that is secreted by the islets of Langerhans within the endocrine pancreas. The beta-cells of the pancreas are glucose sensors. When glucose levels are high, insulin will be released to bind on the insulin receptor on peripheral tissues to promote GLUT4 translocation into the membrane and uptake of glucose [7]. Constant high glucose levels will initiate hyperinsulinemia and resistance of peripheral tissues to insulin, resulting in hyperglycemia. In this context, it is worth mentioning that cancer cells, including non-solid as well as solid malignancies, facilitate glucose uptake by the GLUT1 transporter, which is encoded by the SLC2A1 gene (solute carrier family 2 member 1) [6]. Importantly, the GLUT1 transporter is independent of insulin, enabling cancer cells to substantially uptake glucose, especially in the context of hyperglycemia (e.g., untreated or insufficiently treated diabetes) [8,9]. Nevertheless, insulin appears to impact tumor growth in certain contexts, which is driven by the PI3K (Phosphoinositide 3-kinase) and mTOR (mammalian target of rapamycin) signaling pathway [10].

In general, there are several essential functions of metabolism in cancer cells. Fuel sources (glucose, fatty acids, and amino acids) are utilized for the generation of energy rich phosphates, ATP (adenosine triphosphate), which occurs in the cytoplasm as well as within the mitochondria [11,12,13,14]. In the cytosol, this is an oxidative process as part of glycolysis that generates two ATPs along with the production of reduction equivalents in the form of NADH_2_ (nicotinamide adenine dinucleotide), which can either be oxidized in the lactate dehydrogenase reaction (cytosol) or through complex I of the respiratory chain [6]. Commonly, solid tumors develop areas of hypoxia as they outgrow the oxygen as well as nutrient supply accompanied by activation of the kinase, AMPK (adenosine monophosphate-activated protein kinase) [15]. In this context, anaerobic glycolysis (without oxygen) remains the most viable option to generate ATP, which is accompanied by the accumulation of lactate [6]. While lactate production acidifies the environment, its build up is coupled to the regeneration of NADH to NAD, which is necessary for the glycolytic flux [6].

Glucose, which is critical for biosynthesis, is converted to intermediates. The glucose derivative, glucose-6-phosphate, may be used to promote the pentose phosphate pathway to yield ribulose-5-phosphate that is required for nucleotide synthesis (e.g., building and replication of DNA) (Figure 1). The hexosamine biosynthesis pathway is critical for glycosylation of proteins, which is a post translational modification that enables proper protein function (Figure 1) [16]. In this regard, recent evidence suggests that the hexosamine biosynthesis pathway affects the levels and activity of CDK5 (cyclin dependent kinase 5) in model systems of GBM. In turn, CDK5 modulates acetate metabolism through phosphorylation of ACSS2 (S267) (acyl-CoA synthetase short chain family member 2) and thereby coordinates lipid synthesis [16]. In addition, acetate may serve as a significant source through the enzyme, ACSS2 [16]. 

Cancer cells rely on a constant uptake and metabolization of glucose and utilize glucose carbons for biosynthesis of nucleotides, lipids, and other molecules. To ensure that this process occurs in a highly efficient and swift manner, cancer cells need to readily regenerate the reduction equivalent NADH_2_, which may take place either in the mitochondria or in the cytosol. In the mitochondria, complex I restores NAD, whereas in the cytosol, LDHA (lactate dehydrogenase A) regenerates its levels [6]. LDHA-mediated NAD production appears to be more efficient due to the fact that NAD requires a shuttle system (e.g., the malate-aspartate or the glycerol-3-phosphate shuttles) to deliver its electrons to the respiratory chain. Therefore, the readily conversion of glucose to lactate ensures sufficient cytosolic oxidative power to facilitate glycolysis. It should be noted that, in general, lactate production during glycolysis is a feature of hypoxia since oxygen is a perquisite of NADH_2_/FADH_2_ regeneration during the process of cellular respiration. However, as indicated, tumor cells produce lactate even under physiological oxygen levels, which is referred as the Warburg effect named after a German scientist who coined this term in the last century [17,18]. The vast amounts of lactate are affecting the microenvironment in multiple ways, favoring tumor progression (e.g., growth, angiogenesis, and metastasis). The present literature also suggests that high lactate levels may correlate with a worse prognosis [19,20,21]. In this regard, it has been demonstrated that lactate may act in a manner similar to a cytokine by binding to membranous receptors [22]. However, another more intriguing aspect about lactate is its potential role as a fuel source, which is derived from the idea that in certain tumor types it has been shown that lactate may impact cell survival [23]. For instance, the substrate for histone acetyltransferases, acetyl-CoA, which is derived from lactate, modulated the post-translational modifications of histone H3 such as H3K9ac, H3K14ac, and H3K27ac [23].

Cancer cells depend on their ability to detoxify reactive oxygen species. This process appears to be particularly critical in metastasis, which depends on cellular respiration and oxidative phosphorylation [24]. To block ROS (reactive oxygen species) formation, glutathione is essential, which is a tripeptide that is synthesized from glutamate, cysteine, and glycine [6]. The functional relevant amino acid is cysteine, which contains the necessary thiol groups [11,25]. In addition, the pentose phosphate pathway is a major source of NADPH_2_, which carries reducing power as well. Aside from its role in ROS antagonization, it is critical for biosynthesis of fatty acids and cholesterol, respectively [26,27].

## 3. Major Metabolism Pathways in GBM Model Systems upon Therapy

### 3.1. Metabolic Reprogramming Related to Glycolysis

#### 3.1.1. Glycolysis in the Context of Different Microenvironments

Glycolysis is the break-down of glucose starting in the cytosol with the phosphorylation of HK2 (glucose by hexokinase II) (Figure 1). This process is critical since it constitutes an irreversible reaction. Certain tissues, such as renal and hepatocyte cells, express glucose-6-phosphatase, which would allow the reverse reaction from glucose-6-phosphate [6]. Through a series of reactions, glucose is metabolized to molecules of pyruvate, which contains three carbons [13,17,28]. Depending on the oxygen tension, non-neoplastic cells transfer pyruvate either to the mitochondria or alternatively convert it to lactate in the LDHA reaction (Figure 1). Under non-hypoxic conditions normal cells convert pyruvate to acetyl-CoA in the pyruvate dehydrogenase reaction, requiring thiamine pyrophosphate (also referred to as vitamin B1, which oftentimes is found to be deficient in patients suffering from alcohol abuse) [6]. The remaining two carbons of acetyl-CoA react with citrate and the carbons are oxidized in the TCA-cycle. During these multi-step oxidation reactions there is a significant production of reduction equivalents in the form of NADH_2_ and FADH_2_ (flavin adenine dinucleotide), respectively. In turn, these will be oxidized by either complex I or complex II of the respiratory chain, creating an equivalent of two or three ATPs, respectively. 

In malignant cells glucose is mainly converted to lactate as well as routed to either the pentose phosphate pathway, the serine/glycine pathway, or the hexosamine biosynthesis pathway (Figure 1) [26,27,29,30,31,32,33,34]. To ensure rapid progression of glycolysis, cancer cells rely on the activity of LDHA to regenerate NAD, which provides them with a growth advantage [8,9]. Still, parts of pyruvate or potentially alternatively lactate (when one would consider this formerly coined waste product as a fuel source) enter the mitochondria and are converted to acetyl-CoA, which reacts to citrate. In this regard, it was demonstrated that glioma cancer stem cells are affected by the pyruvate dehydrogenase kinase inhibitor, dichloroacetate (DCA), to shift their metabolism towards cellular respiration, whereas this effect was not observed in neural stem cells [35]. Citrate is capable of crossing the mitochondrial membrane to the cytosol and in turn may be cleaved to acetyl-CoA by the ATP citrate lyase enzyme, encoded by ACLY (ATP citrate synthase), which may fuel the biosynthesis of fatty acids and cholesterol or affect histone compaction (Figure 1) [8,11,12,13]. 

#### 3.1.2. Drug-Induced Reprogramming of Glycolysis in GBM Model Systems

Glycolysis is regulated at multiple levels, e.g., post-translational modifications of enzymes, transcriptional regulation, and others [36,37]. In this context, c-Myc and HIF1A (hypoxia-inducible factor 1-alpha) are likely the most relevant transcription factors to control glycolysis by regulation of key enzymes (e.g., HK2 and LDHA) and transporters (GLUT1) [38,39,40,41,42,43]. For instance, mTORC2 (mTOR complex 2) was shown to affect and regulate glycolysis via c-Myc through a complex mechanism, involving histone deacetylases as well as FoxO1 (forkhead box O1) and FoxO3 (forkhead box O3) in GBM cells [41]. Recent research has shown that HDAC1 (histone deacetylase 1) and HDAC2 (histone deacetylase 2) appear to regulate glycolysis by driving c-Myc expression [44]. In this context, HDAC1/2 inhibitors, such as romidepsine and panobinostat, blocked c-Myc expression by disrupting the c-Myc super-enhancer in GBM cells (Table 1). HDAC-inhibitor-mediated loss of glycolytic enzymes was accompanied by a reduction of glycolytic metabolites and reduced glycolytic activity as assessed by extracellular flux analysis [44]. Akin to observations related to HDAC inhibitors, recent findings suggest that aurora kinase A is involved in glycolysis regulation [45]. Interference with its function results in a reduction of GBM growth. Remarkably, it was found that loss of function of aurora kinase A resulted in a more substantial loss of cellular viability under conventional glucose conditions (between 5 mM and 25 mM). Galactose is an alternate sugar that may be supplemented in cell culture media in lieu of glucose, which generally renders cells more dependent on cellular respiration [45,46]. Once the glucose is replaced with galactose in growth media, GBM cells were substantially more resistant to the cytotoxic effects of the aurora kinase A inhibitor, alisertib, which strongly suggest that aurora kinase A inhibition is more potent in highly glycolytic contexts (Table 1). Conversely, GBM cells that rely more on oxidative energy metabolism appear to be more likely to escape from aurora kinase A inhibitor therapy. Mechanistically, aurora kinase A interacts with c-Myc and loss of aurora kinase A activity drives a proteasomal mediated reduction of c-Myc in GBM cells, which is associated with a reduction of GLUT1, HK2, and LDHA [45]. Phosphorylation of c-Myc is known to be mediated by GSK3β (Glycogen synthase kinase-3 beta) [47,48]. Loss of aurora kinase A activity facilitated the activation of GSK3β, which drove c-Myc degradation as demonstrated by loss function studies related to GSK3β [45]. 

### 3.2. Metabolic Reprogramming of the Pentose Phosphate Pathway

Following activation by hexokinase, glucose-6-phosphate may be metabolized in the pentose phosphate pathway (PPP) [26,27]. The PPP is known for its role in an inherited disease that is characterized by hemolysis due to deficiency in the first enzyme of this pathway, glucose-6-phosphate dehydrogenase. The reason for the “hemolysis phenotype” relates to the depletion of NADPH_2_ in the erythrocytes, serving as an ROS scavenger. Consistently, this part of the PPP is referred to as the oxidative part since it oxidizes glucose and in turn reduces NADP to NADPH, creating “biosynthetic reducing power for the cancer cells to grow” [8,14,49,50]. Moreover, the end product of the pathway, ribulose-5-phosphate, is the precursor of ribose, which is part of nucleotide and amino acid biosynthesis. Under hypoxic conditions, HIF1A drives the up-regulation of BCAT1 (inhibited through gabapentin) and LAT1 (L-type amino acid transporter 1) to facilitate the reaction of branched chain amino acids, e.g., leucine, isoleucine, and valine, to glutamate [51].

Recently, it has been shown that glioblastomas with active EGFR signaling are particularly reliant on the glucose-6-phosphate dehydrogenase [52]. Upon activation of EGFR signaling, the SRC kinase (Fyn) phosphorylates a tyrosine residue of glucose-6-phosphate dehydrogenase (pY481), resulting in an increase of its activity which is mediated through enhanced binding of oxidized NADP with glucose-6-phosphate dehydrogenase [52]. The non-oxidative pathway connects ribose-5-phosphate with the glycolytic pathway through two key enzymes, called transketolase and transaldolase [53]. Notably, the enzyme human telomerase reverse transcriptase (hTERT) has been shown to affect the activity of the PPP by regulation of the expression of both transketolase and glucose-6-phosphate dehydrogenase [53]. It was also shown that constitutively active EGFR signaling drives lipid synthesis in glioblastoma models through a PI3K/Akt/SREBP-1 signaling axis. In turn, interference with SREBP-1 or FASN kills GBM cells reliant on this pathway [54]. In addition, loss of function of HDAC1/2 by panobinostat reduced the metabolization of glucose in the PPP, resulting in down-regulation of ribose-5-phosphate as well as modulation of the NADPH_2_/NADP ratio [44] (Table 1).

### 3.3. The Tricarboxylic Acid (TCA) Cycle and Oxidative Phosphorylation (OXPHOS)

#### 3.3.1. Basic Functions of the TCA Cycle and Cellular Respiration

The TCA-cycle bears both catabolic as well as anabolic functions (Figure 2). With regards to its catabolic functions, pyruvate is oxidized to acetyl-CoA in a manner dependent on thiamine pyrophosphate [6]. In turn, acetyl-CoA reacts with oxaloacetate to form citrate. In addition, ketone bodies as well as fatty acids are oxidized to acetyl-CoA (not itself part of the TCA-cycle), which in turn reacts with the TCA-cycle [11,55]. Glutamine is converted to glutamate and oxidized to α-ketoglutarate. One turn of the TCA-cycle generates NADH_2_, FADH_2,_ and GTP (Guanosine-5’-triphosphate). It is important to recognize that acetyl-CoA is not anaplerotic, whereas in contrast, the pyruvate carboxylase reaction that creates oxaloacetate from pyruvate is anaplerotic. The pyruvate carboxylase reaction becomes critical in contexts in which a significant amount of acetyl-CoA is produced (e.g., during enhanced beta oxidation). Another anaplerotic reaction is the formation of alpha-ketoglutarate from glutamate, which is derived from glutamine (Figure 2) [8]. The oxidation of NADH_2_ occurs at complex I of the respiratory chain in the inner mitochondrial membrane. The electrons are passed to complex IV (via complex III) to reduce molecular oxygen to water [8,28]. The oxidation reaction yields energy that is used by complex I, II, and III to pump protons into the mitochondrial intermembrane space. This electrochemical gradient is used to fuel the creation of ATP from ADP and phosphate at complex V (ATP-synthase) [6]. 

#### 3.3.2. Drug Compounds That Interfere with the TCA-Cycle and OXPHOS

While some GBM cells appear to be more reliant on glycolysis, there are others that are dependent on oxidative metabolism to a larger degree [56]. In this context, it is noteworthy that there are a number of different metabolic inhibitors that interfere either with the TCA-cycle, OXPHOS, or both. In this regard, there is a compound that targets both PDH (Pyruvate dehydrogenase) and alpha-ketoglutarate dehydrogenase called devimistat or CPI-613 [9,57,58] (Table 1). Recent research suggests that CPI-613 kills GBM cells more potently under conditions that favor oxidative metabolism by binding to PDHA1 [23]. CPI-613 reduced the levels of TCA-cycle metabolites in vivo and extended overall survival in orthotopic model systems of GBM, which was accompanied by an induction of cell death and reduction of proliferation [23].

An additional family of drug compounds that affect both the TCA-cycle and cellular respiration are imipridones. The first one of these compounds was initially called TIC10 (TRAIL inducing compound 10) since it was identified in a screen of compounds that induce TRAIL (TNF-related apoptosis-inducing ligand) expression [40,59,60,61]. TRAIL was considered the “holy grail” in cancer therapy since it specifically induced apoptosis in cancer cells, but not in normal cells. These effects were mediated mainly by two receptors, DR4 (death receptor 4) and DR5 (death receptor 5), and the literature suggests that aside from up-regulation of TRAIL, TIC10 facilitated the expression of DR5 as well [61]. The compound was later renamed and referred to as ONC201. Further research clarified the mechanisms by which imipridones, which also include the more potent derivatives ONC206 and ONC212, elicit reduction of cellular viability in cancer cells [40,60,62,63]. Viability studies, involving GBM cells, suggested that low glucose conditions sensitized for imipridone-mediated cell death. Similarly, the glucose analogue, 2-DG (2-Deoxy-D-glucose), when combined with ONC201, induced synergistic reduction of cellular viability in GBM cells [62]. Moreover, ONC201, ONC206, and ONC212 substantially reduced the oxygen consumption rate in GBM cells, suggesting that imipridones in part reduce cancer cell viability by suppression of cellular respiration and ATP-coupled respiration [40,63]. Imipridone-mediated suppression of OXPHOS resulted in energy deprivation which is accompanied by an increase of stress-response-related transcription factors, e.g., ATF4. Further research clarified the mechanisms by which imipridone suppresses OXPHOS, which involves the activation of the mitochondrial caseinolytic protease CLPP (caseinolytic mitochondrial matrix peptidase proteolytic subunit) protease [63,64] (Table 1). The imipridone binds to CLPP and activates it, resulting in the depletion of respiratory complexes, including NDUFA12 (part of complex I) (NADH: ubiquinone oxidoreductase subunit A12), SDHA (succinate dehydrogenase complex flavoprotein subunit A), SDHB (succinate dehydrogenase complex flavoprotein subunit B) as well as proteins related to both complex IV and V [63]. Others have shown that blockage of complex I through metformin in combination with 2-DG suppressed the growth of GBM neurospheres in vitro and in vivo, consistent with the concept that upon loss of glycolytic function, GBM cells rely heavily on cellular respiration, and in turn, interference with complex I becomes readily toxic for the cells [65,66]. 

Recently an interesting link was established between intracellular cholesterol levels and activity of cellular respiration in GBM cells. Liver X receptor (LXR) agonists are known for their ability to deplete cells of cholesterol (via induction of the ABCA1 (ATP binding cassette subfamily A member 1) transporter and through reduction of the LDL-receptor) and induce cell death [67,68,69,70]. It was shown that the LXR agonist, LXR623, reduces cell death in a manner dependent on its ability to suppress cellular respiration (Table 1). Mechanistically, cholesterol appears to be necessary to stabilize the protein levels of the respiratory complexes, and consistent treatment of GBM cells with LXR623 led to a suppression of several electron transport chain proteins, resulting in a substantial reduction of the oxygen consumption rate of GBM cells [69]. It is notable that these effects on metabolism appear to preferentially occur in GBM cells over normal cells. Like other blockers of cellular respiration, low levels of glucose or culturing GBM cells in galactose resulted in enhanced killing potency by LXR623. Akin to imipridones, the activation of LXR is associated with an increase of ATF4 (activating transcription factor 4), which itself regulates Noxa expression, a pro-apoptotic Bcl-2 family member that is known to enhance susceptibility towards BH3-mimetics [69]. A further modality to block tumor cell respiration is the antagonization of mitochondrial Hsp90 (heat shock protein 90) and the homologue TRAP1 (tumor necrosis factor receptor-associated protein 1). Gamitrinib inhibits mitochondrial Hsp90 and recently this class of compounds has entered clinical testing [70] (Table 1). A single treatment with gamitrinib reduced growth of multiple different tumor types, including hematological as well as solid malignancies. Originally, the mechanism of action of these compounds was linked to an unfolded stress response linked to hexokinase inhibition [71]. Later, it was noted that most likely the primary mechanism of these compounds is the blockage of mitochondrial respiration, in part mediated through the destabilization of SDHB [69].

#### 3.3.3. Loss of Function of HDAC1/2, AURKA and MET Drives Oxidative Energy Metabolism

As mentioned earlier HDAC1/2 appear to regulate carbohydrate metabolism and upon their inhibition glycolysis is reduced, but at the same time respiration is increased in GBM cells [44]. The enhanced activity of cellular respiration is fueled by an increase in beta oxidation and elevated anaplerosis via the pyruvate carboxylase reaction (Figure 2). Mechanistically, HDAC inhibitions decreased the levels of c-Myc. Chromatin immunoprecipitation experiments highlighted that c-Myc binds to the promoter region of the transcription factor PGC1A (peroxisome proliferator-activated receptor gamma coactivator 1-alpha), which is a driver of cellular respiration [44,72]. Within the PGC1A promoter, c-Myc appears to act as a repressor since loss of c-Myc upregulates PGC1A transcription and protein expression. In turn, HDAC inhibitors facilitated an increase of PGC1A, which facilitated enhanced oxygen consumption rate and drove survival of GBM cells since knock-down of PGC1A abrogated these effects [44]. To translate these findings into a potentially clinical scenario, panobinostat was combined with the fatty acid oxidation inhibitor, etomoxir, which has previously been tested in clinical trials [73,74]. The combination treatment synergistically reduced the cellular viability of all kinds of GBM cultures, including neurospheres, PDX-derived lines, as well as established GBM cells. The reduction of viability was linked in part to a cell death with apoptotic features and consistently anti-apoptotic Bcl-2 (B-cell lymphoma-2) family members were reduced [44]. In vivo, etomoxir along with panobinostat extended animal survival in a PDX-derived orthotopic GBM model, suggesting potential translational relevance of the findings. Similar observations were made in the setting of inhibition of aurora kinase A [44,45]. In this scenario, loss of function of aurora kinase A yielded c-Myc degradation with concomitant increase of PGC1A and activation of fatty acid oxidation, which was coupled with a suppression of glycolysis as well [45]. Consistently, the combination of alisertib along with etomoxir extended animal survival in orthotopic PDX models of GBM. Moreover, alisertib or loss of function of aurora kinase A along with gamitrinib or inhibitors of OXPHOS, synergistically reduced the viability of GBM cells [45]. 

Since imipridones primarily appear to block cellular respiration to elicit anti-tumor activity, it was tempting to speculate whether combining ONC201, ONC206, and ONC212 along with HDAC1/2 inhibitors would result in synergistic killing of GBM cells. Indeed, recent findings confirm that this drug combination is more effective in killing GBM cells than each compound by itself [63]. The combination of imipridone and HDAC inhibitors elicited cell death with apoptotic features which was partially rescued by pan-caspase inhibitors and involved the modulation of anti-apoptotic Bcl-2 family members [63]. Rescue experiments with either Mcl-1 (myeloid cell leukemia-1) or Bcl-xL (B-cell lymphoma-extra large) confirmed their involvement in the death induced by the combination treatment, in keeping with features of the intrinsic apoptotic pathway. The combination treatment was strictly dependent on CLPP activation, as demonstrated by several loss function experiments. In addition, based on a high-throughput drug screen involving gamitrinib (G-TPP), we discovered that blockage of TRAP1 along with HDAC1/2 is synthetically lethal in GBM model systems, which was confirmed in vivo as well [75,76]. 

A further example of an increase of OXPHOS and fatty acid oxidation is represented by inhibition of the MET kinase (MET proto-oncogene, receptor tyrosine kinase). The MET gene is amplified in a fraction of GBMs and therefore this protein constitutes a viable target [77]. However, as in other contexts, interference with MET was not durable in GBM, raising the question of how resistance could occur so rapidly. One possible explanation may be metabolic reprogramming following treatment with the MET inhibitor, crizotinib (Table 1). Indeed, crizotinib treatment led to an up-regulation of acyl-carnitines and a transcriptional signature of fatty acid oxidation [77]. Carbon tracing experiments and extracellular flux analysis confirmed the increase of oxidative energy metabolism fueled in part by palmitic acid [77]. In the context of crizotinib, it was noted that glucose was critical to provide carbons to the TCA cycle such that citrate was substantially more labeled by glucose carbons following crizotinib exposure. In addition, the extracellular acidification rate was elevated, suggesting an increase in glycolysis [77]. This metabolic situation induced by crizotinib appears to be different than in the context of HDAC and aurora kinase A inhibitors. While for all these compounds it appears to be common to activate anaplerosis through the pyruvate carboxylase reaction, crizotinib-related metabolic reprogramming appears to have a glycolytic component with even increases of lactate [44,45,77]. Consistently, crizotinib-exposed GBM cells displayed enhanced phosphorylation of PDHA. Related to recent work, it may be conceivable that the crizotinib-mediated increase in lactate might be used to fuel anaplerosis in the TCA-cycle by re-uptake of lactate to the mitochondria and metabolization through LDHB. The metabolic advantage of this scenario would be that, through this pathway, the rapid regeneration of NAD in the cytosol is ensured, providing sustained and uninterrupted glucose metabolization. It is interesting to note that in all instances of treatment with crizotinib, alisertib, panobinostat, and romidepsin, glutamine oxidation was substantially suppressed, confirming the importance of fatty acid oxidation in these settings [44,45,77]. A recent report also suggested that the standard of care for GBM, temozolomide, appears to have an impact on fatty acid metabolism since it facilitates the uptake of fatty acids, potentially used for fatty acid oxidation (Figure 2) [78]. 

## 4. A Novel Fuel for the TCA-Cycle in GBM

Lactate is the end product of glycolysis and thus far was mainly viewed as a carbon source in the context of gluconeogenesis in the liver. Recent data in non-small cell lung cancer suggested that these tumors are actually capable of metabolizing lactate and that in fact lactate carbons may be the preferential carbon source over glucose to contribute to the TCA-cycle [19,23,79]. With regards to lung carcinoma, these tumors are usually embedded in an environment that has ample oxygen, thus potentially enabling efficient oxidative metabolization. Lactate appears to have a role in facilitating metastasis in model systems of melanoma [80]. Melanomas that utilized lactate efficiently displayed a higher rate of metastasis. Lactate uptake was facilitated by the transport protein, MCT1 (Monocarboxylate transporter 1) [36,81,82]. Consistently, loss of function of MCT1 diminished metastasis, which was associated with an increase in ROS likely caused by a suppression of the NADPH_2_ generating PPP [80]. GBMs and other solid tumors are known to have areas of low glucose levels, suggesting that either alternate fuel sources may be utilized or that uptake of glucose may be enhanced. Regarding the latter, an earlier paper suggested this scenario in the context of brain tumor initiating cells (BTICs). When these cells were exposed to low glucose levels they up-regulated a glucose transporter (GLUT3) to facilitate glucose uptake. High levels of GLUT3 correlated with a worse outcome in GBM and knockdown of GLUT3 interfered with BTIC growth in orthotopic GBM xenografts in mice [83].

Recently, it was demonstrated that lactate is being taken up by various GBMs of different genetic and molecular backgrounds, including lines with mutated R132H IDH1 (Isocitrate dehydrogenase 1), which appeared to be mediated by the membranous transport protein MCT1 (Figure 3). MCT1 itself is significantly up-regulated in glioblastoma tissue when compared with normal brains (TCGA database). Most relevantly, lactate was able to rescue a large array of different GBM cells from glucose deprivation in culture, which was reliant on functional cellular respiration since blockage of complex I through IACS-010759 reversed the rescue through lactate almost entirely (Table 1). Similar observations were made with oligomycin, suggesting that both respiration as well as respiration-linked ATP production are involved in the process [23] (Table 1). Consistently, U-13C-lactate labeled the TCA-cycle and related amino acids. Not surprisingly, acetyl-CoA was heavily labeled as well. Lactate-labeled acetyl-CoA served as a substrate for histone acetyltransferases to modify histone proteins at lysine residues, such as H3K27ac, H3K9ac, and H3K14ac. Mass spectrometry data from histones confirms that histone acetyl-residues contained carbons from lactate [23]. In turn, the activating histone marks localized to promoters and enhancers that are related to cellular respiration. Cytosolic acetyl-CoA is produced from lactate in a manner dependent on PDHA1, citrate synthase, and the ATP citrate lyase (Figure 3). Consistently, lactate-mediated survival of GBM cells was blocked by loss of function of PDHA1 (pyruvate dehydrogenase E1 subunit alpha 1) and ACLY, which was also associated with a reduction of histone acetylation.

## 5. Conclusions

Thus far, GBMs remain tumors with a bad prognosis. Still, surgery, temozolomide, and radiation remain the “bread and butter” in terms of therapy despite the fact that it only provides a limited therapeutic benefit. Despite targetable vulnerabilities, such as EGFR or MET, these malignancies readily escape from therapy. In lung adenocarcinoma, targeting these molecules is clearly more effective as demonstrated by clinical trials [84]. The reasons for therapeutic failure may be related to rapid metabolic reprogramming upon therapy. In this review, we summarized various examples of targeted therapies in preclinical models of GBM that appear to exert a major impact on GBM energy metabolism by affecting glucose, glutamine, and fatty acid utilization. Most notably, it seems that drug-induced metabolic reprogramming mainly enhances fatty acid oxidation, while glutamine oxidation is reduced. This observation may be critical since it would allow the targeting of these alterations upfront, thereby preventing potential resistance to therapy. In this context, inhibitors of fatty acid oxidation are available for clinical use and may be combined with the standard of care for targeted drug therapy. Underappreciated fuel sources and their related pathways may also be worthwhile as therapeutic targets.

## Figures and Tables

**Figure 1 cells-11-02956-f001:**
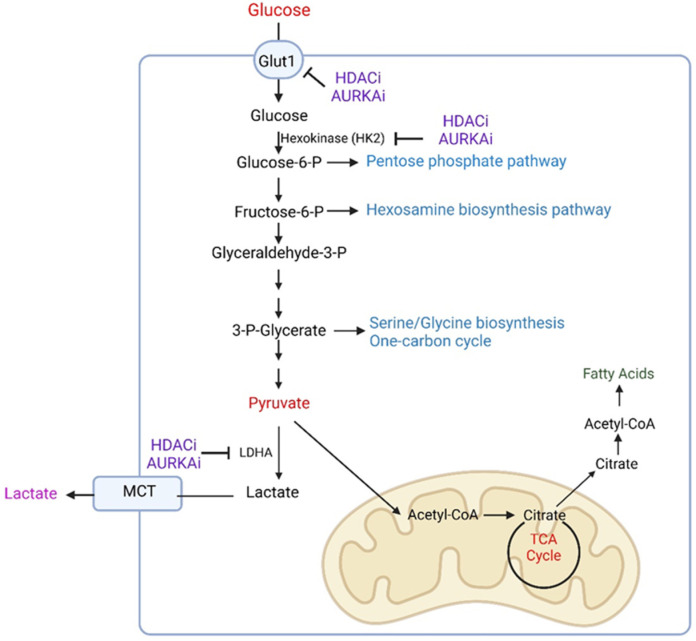
Glycolysis and related pathways in GBM cells. Glucose enters GBM cells via GLUT1 in a manner independent of insulin. Upon entry, glucose is phosphorylated to glucose-6-phosphate and may be processed further in the pentose phosphate pathway or in glycolysis. Glycolysis feeds and communicates with the hexosamine biosynthesis pathway (at the level of fructose-6-phosphate) as well as the serine/glycine pathway (originating from 3-phospho-glycerate). The final product of glycolysis is pyruvate, which may subsequently convert to lactate, resulting in regeneration of NADH to NAD, facilitating rapid glycolytic flux. Lactate is removed from the cell through either MCT1 or MCT4. Alternatively, pyruvate is converted to acetyl-CoA, which in turn reacts to citrate. Citrate may be either processed in the TCA-cycle or used to give rise to cytosolic acetyl-CoA, which is used for fatty acid synthesis. HDAC and aurora kinase A inhibitors block glycolysis by suppression of GLUT1, hexokinase II, and LDHA.

**Figure 2 cells-11-02956-f002:**
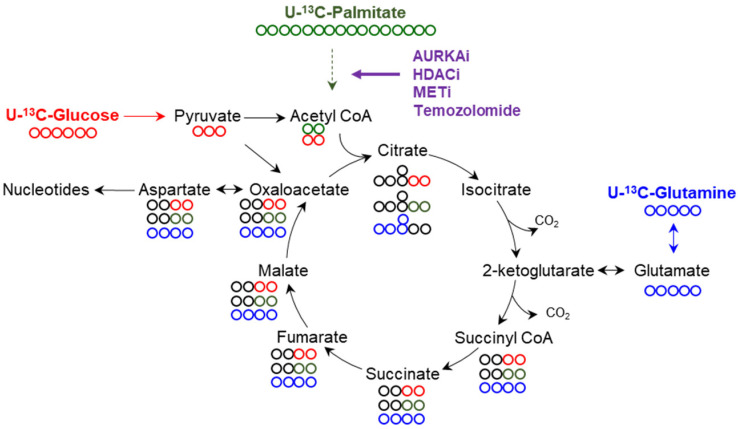
The tricarboxylic acid cycle is fueled by several major sources including glucose, glutamine, and palmitic acid. During glycolysis, glucose (containing six carbons) reacts to 2 molecules of pyruvate. Pyruvate reacts either to oxaloacetate (pyruvate carboxylase reaction) or is oxidized to acetyl-CoA. Acetyl-CoA reacts with oxaloacetate to citrate. Palmitic acid (U-C13) is oxidized (beta-oxidation) and the final product is acetyl-CoA as well. It is noteworthy that both uniformly labeled glucose and palmitic acid result in the m+2 citrate isotopologue (originating from m+2 acetyl-CoA). The carboxylation reaction (pyruvate to oxaloacetate) results in m+3 labeled citrate (not shown). U-13C-Glutamine gives rise to the m+5 alpha-ketoglutarate isotopologue (anaplerotic reaction) and in turn is oxidized in the TCA-cycle. HDAC, aurora kinase A, MET inhibitors, and temozolomide facilitate beta oxidation, enhance anaplerosis through the pyruvate carboxylase reaction, and dampen glutamine oxidation.

**Figure 3 cells-11-02956-f003:**
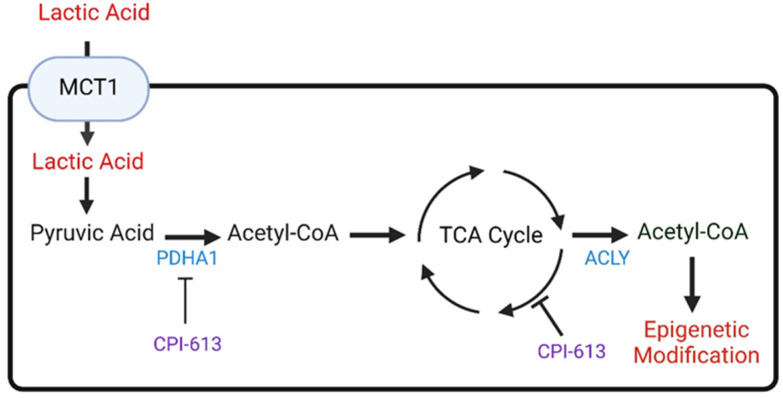
Lactic acid is a substrate of the TCA-cycle and involved in acetyl-CoA production. GBM cells take up lactic acid through the membranous transporter MCT1 and convert it to pyruvic acid. Pyruvic acid is subjected to oxidative decarboxylation by PDHA1, resulting in the release of CO_2_ and production of acetyl-CoA. Acetyl-CoA is converted into citrate in the TCA cycle and released to the cytosol. The ACLY reaction produces acetyl-CoA that in turn serves as a substrate for histone acetyltransferases that modify histone H3 and H4, resulting in enhanced accessibility of chromatin and transcription of genes facilitating proliferation. CPI613 interferes with lactate metabolism by blocking PDHA and interference with the TCA-cycle. The image is adapted from [23].

**Table 1 cells-11-02956-t001:** A summary of therapeutic drugs with their specific inhibition/activation targets and the metabolic pathway affected by drug treatment.

Drugs	Inhibit/Activate Target	Metabolic Pathways
Panobinostat	pan-HDAC	Glycolysis
Romidepsin	HDAC1/2	Glycolysis
Alisertib	Aurora kinase A	Glycolysis
Devimistat (CPI-613)	Pyruvate dehydrogenase	TCA-cycle
Imipridones (ONC201, ONC206, ONC212)	CLPP	OXPHOS Cellular respiration
LXR623	LXRβ	OXPHOS
Gamitrinib (G-TPP)	Hsp90/TRAP1	OXPHOS
Crizotinib	MET kinase	OXPHOS
Metformin	Complex I of respiratory chain	OXPHOS
IACS-010759	Complex I of respiratory chain	OXPHOS
Oligomycin	ATP synthase–Complex V	OXPHOS
Etomoxir	CPT1A	Fatty acid oxidation

## Data Availability

Not applicable.

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
