# Peer review of "Therapeutic Drug-Induced Metabolic Reprogramming in Glioblastoma"

_cells, 2022, doi:10.3390/cells11192956_

Round 1

Reviewer 1 Report

This is a review article by Nguyen et al., highlighting recent research related to therapy-induced metabolic reprogramming in glioblastoma (GBM). This manuscript is clear, well written, and nicely summarizes the latest research findings pertinent to this important topic. Below are a few suggestions to increase the clarity and breadth of this review:

- I suggest revisiting the title which does not accurately reflect the main topic of this review, i.e. therapy/drug-induced metabolic reprogramming in GBM.

- Paragraph 2 (page 2) is a bit confusing or misplaced. Although the title mentions metabolism in general, this paragraph seems to primarily focus on glucose metabolism. I suggest either changing the title or preferably moving this paragraph somewhere else.

- Paragraph 3.1 is long and a bit complex to comprehend as currently presented. It would be better to divide it into different subparagraphs and with appropriate subheadings (similar to 3.2).

- Several tyrosine kinases such as EGFR have been linked to de novo lipid synthesis. While this might not be the focus of this manuscript, I recommend adding a few sentences related to oncogenic signaling and lipogenesis as well.

- I recommend adding a table summarizing some of the key compounds/drugs discussed in this review (kinase inhibitors, HDAC inhibitors, and others), their intended target, and the metabolic pathways affected by such drugs.

- Please add references to statements in lines 325-327

- Minor comment: Please define all the acronyms used in the manuscript after the first use; for ex. TRAIL (line 251), DR4 (line 254) etc.

Author Response

This is a review article by Nguyen et al., highlighting recent research related to therapy-induced metabolic reprogramming in glioblastoma (GBM). This manuscript is clear, well written, and nicely summarizes the latest research findings pertinent to this important topic. Below are a few suggestions to increase the clarity and breadth of this review:

Response: Thank you so much for your thorough review.

- I suggest revisiting the title which does not accurately reflect the main topic of this review, i.e. therapy/drug-induced metabolic reprogramming in GBM.

Response: As suggested, we changed the title to “therapeutic drug-induced metabolic reprogramming in GBM”.

- Paragraph 2 (page 2) is a bit confusing or misplaced. Although the title mentions metabolism in general, this paragraph seems to primarily focus on glucose metabolism. I suggest either changing the title or preferably moving this paragraph somewhere else.

 Response: As suggested we changed the title of this paragraph to “Glucose metabolism as a central component of GBM and tumor growth”.

- Paragraph 3.1 is long and a bit complex to comprehend as currently presented. It would be better to divide it into different subparagraphs and with appropriate subheadings (similar to 3.2).

Response: We followed these suggestions and added two additional subheadings accordingly (page 3-4).

- Several tyrosine kinases such as EGFR have been linked to de novo lipid synthesis. While this might not be the focus of this manuscript, I recommend adding a few sentences related to oncogenic signaling and lipogenesis as well.

Response: We have expanded on this topic and added the related references to the manuscript (page 6, line 227-229)

- I recommend adding a table summarizing some of the key compounds/drugs discussed in this review (kinase inhibitors, HDAC inhibitors, and others), their intended target, and the metabolic pathways affected by such drugs.

Response: As requested we added a table to clarify this point (Table 1, page 6).

- Please add references to statements in lines 325-327

 Response: we added the reference to the statement in lines 325-327

- Minor comment: Please define all the acronyms used in the manuscript after the first use; for ex. TRAIL (line 251), DR4 (line 254) etc.

Response: As requested, we defined all the relevant acronyms accordingly.

Reviewer 2 Report

This is the great work of current review. The authors review recent key findings in the literature that highlight that GBM cells substantially reprogram their metabolism upon therapy. These studies suggest that blocking glycolysis will yield a concomitant reactivation of oxidative energy pathways and most dominantly beta-oxidation of fatty acids. However, there are some issues need to declare.

 1. Do the GBM cells and GBM neural stem cells play the same role in glucose related pathway?

2. In general, there are several essential functions of metabolism in cancer cells. Fuel 66 sources (glucose, fatty acids, and amino acids) are utilized for the generation of energy 67 rich phosphates: adenosine triphosphate (ATP), which occurs in the cytoplasm as well as 68 within mitochondria The GBM take energy from amino acid by LAT1/LAT2 massively. What does the amino acid play the role in glucose/TCA-cycle related pathway?

Author Response

This is the great work of current review. The authors review recent key findings in the literature that highlight that GBM cells substantially reprogram their metabolism upon therapy. These studies suggest that blocking glycolysis will yield a concomitant reactivation of oxidative energy pathways and most dominantly beta-oxidation of fatty acids. However, there are some issues need to declare.

Response: We thank the reviewer for pointing out all these interesting points, which we have considered in our revision.

  1. Do the GBM cells and GBM neural stem cells play the same role in glucose related pathway?

Response: To address this critical point, we have added the following statement to the manuscript. “In this regard, it was demonstrated that glioma cancer stem cells are affected by the pyruvate dehydrogenase kinase inhibitor, dichloroacetate (DCA), to shift their metabolism towards cellular respiration, whereas this effect was not observed in neural stem cells.” (Manuscript, line 153).

  1. In general, there are several essential functions of metabolism in cancer cells. Fuel 66 sources (glucose, fatty acids, and amino acids) are utilized for the generation of energy 67 rich phosphates: adenosine triphosphate (ATP), which occurs in the cytoplasm as well as 68 within mitochondria The GBM take energy from amino acid by LAT1/LAT2 massively. What does the amino acid play the role in glucose/TCA-cyclerelated pathway?

Response: We added a statement related to amino acids in the pentose phosphate pathway (glycolysis pathway). We pointed out that under hypoxic conditions LAT1 is upregulated and that branched chain amino acids are utilized as a fuel source under these conditions in GBM models. A relevant reference was cited as well (manuscript Page 5).  

Reviewer 3 Report

The paper is concisely written, with clear message. The only two comments are to additionally mention that radiotherapy and concomitant temozolomide provides only a modest increase of survival and temozolomide action depicted in a scheme.

Author Response

The paper is concisely written, with clear message. The only two comments are to additionally mention that radiotherapy and concomitant temozolomide provides only a modest increase of survival and temozolomide action depicted in a scheme.

Response: We thank the reviewer for pointing out all these interesting points, which we have considered in our revision (Figure 2 and manuscript page 11, line 470).